Analyses of antioxidant status and nucleotide alterations in genes encoding antioxidant enzymes in patients with benign and malignant thyroid disorders

Ramli Nur Siti Fatimah 1
Mat Junit Sarni 1
Leong Ng Khoon 2
http://orcid.org/0000-0002-8661-3822 Razali Nurhanani 1
Jayapalan Jaime Jacqueline 3
Abdul Aziz Azlina 1 azlina_aziz@um.edu.my
1 Department of Molecular Medicine, Faculty of Medicine, University of Malaya , Kuala Lumpur , Malaysia
2 Department of Surgery, Faculty of Medicine, University of Malaya , Kuala Lumpur , Malaysia
3 University of Malaya Centre for Proteomics Research, University of Malaya , Kuala Lumpur , Malaysia
Zhou Shengtao
Electronic publication date: 2017 Jun 1
Publication date: 2017
Volume: 5
Electronic Location ID: e3365
Received 2016 Oct 26; Accepted 2017 Apr 28
Copyright: © 2017 Ramli et al.
Copyright year: 2017
Copyright holder: Ramli et al.
License: This is an open access article distributed under the terms of the Creative Commons Attribution License, which permits unrestricted use, distribution, reproduction and adaptation in any medium and for any purpose provided that it is properly attributed. For attribution, the original author(s), title, publication source (PeerJ) and either DOI or URL of the article must be cited.
License URL: https://creativecommons.org/licenses/by/4.0/

Keywords: Thyroid disorders, Antioxidants, Antioxidant enzymes, Whole exome sequencing, Single nucleotide polymorphisms, Oxidative stress

Funding: Fundamental Research Grant Scheme FP034-2014A University of Malaya PG115-2014A This work was supported by the Fundamental Research Grant Scheme (FP034-2014A) and the Postgraduate Research Grant, University of Malaya (PG115-2014A). The funders had no role in study design, data collection and analysis, decision to publish, or preparation of the manuscript.

==============================
Background

Synthesis of thyroid hormones and regulation of their metabolism involve free radicals that may affect redox balance in the body. Thyroid disorders causing variations in the levels of thyroid hormones may alter cellular oxidative stress. The aim of this study was to measure the antioxidant activities and biomarkers of oxidative stress in serum and red blood cells (RBC) of patients with benign and malignant thyroid disorders and to investigate if changes in the antioxidant activities in these patients were linked to alterations in genes encoding the antioxidant enzymes.

Methods

Forty-one patients with thyroid disorders from University of Malaya Medical Centre were recruited. They were categorised into four groups: multinodular goitre (MNG) (n = 18), follicular thyroid adenoma (FTA) (n = 7), papillary thyroid cancer (PTC) (n = 10), and follicular thyroid cancer (FTC) (n = 6). Serum and RBC of patients were analysed for antioxidant activities, antioxidant enzymes, and biomarkers of oxidative stress. Alterations in genes encoding the antioxidant enzymes were analysed using whole exome sequencing and PCR–DNA sequencing.

Results

Patients with thyroid disorders had significantly higher serum superoxide dismutase (SOD) and catalase (CAT) activities compared to control, but had lower activities in RBC. There were no significant changes in serum glutathione peroxidase (GPx) activity. Meanwhile, GPx activity in RBC was reduced in PTC and FTC, compared to control and the respective benign groups. Antioxidant activities in serum were decreased in the thyroid disorder groups when compared to the control group. The levels of malondialdehyde (MDA) were elevated in the serum of FTA group when compared to controls, while in the RBC, only the MNG and PTC groups showed higher MDA equivalents than control. Serum reactive oxygen species (ROS) levels in PTC group of both serum and RBC were significantly higher than control group. Whole exome sequencing has resulted in identification of 49 single nucleotide polymorphisms (SNPs) in MNG and PTC patients and their genotypic and allelic frequencies were calculated. Analyses of the relationship between serum enzyme activities and the total SNPs identified in both groups revealed no correlation.

Discussion

Different forms of thyroid disorders influence the levels of antioxidant status in the serum and RBC of these patients, implying varying capability of preventing oxidative stress. A more comprehensive study with a larger target population should be done in order to further evaluate the relationships between antioxidant enzymes gene polymorphisms and thyroid disorders, as well as strengthening the minor evidences provided in literatures.

Introduction

Thyroid hormones play important roles in the body including regulating metabolic rate and oxidative metabolism as well as stimulating growth and protein synthesis. Thyroid gland dysfunction is known to alter lipid profiles, potentially causing conditions such as hypertension, insulin resistance and oxidative stress. During the production of thyroid hormones, reactive oxygen species (ROS) are generated as part of the process. However, under normal redox balance, the ROS are removed by antioxidant systems in the body, hence limiting oxidative damage. On the other hand, certain conditions such as inflammation of the thyroid gland as well as tumour cell proliferation could alter the balance between ROS and antioxidant levels (Cetin et al., 2015) in favour of the former, subsequently leading to oxidative damage (Erdamar et al., 2010). In particular, the utilisation of H2O2 for thyroid hormones synthesis causes the thyroid to be easily exposed to oxidative stress in situation where there is disorder in thyroid hormones production (Karbownik-Lewińska & Kokoszko-Bilska, 2012).

Several studies have shown the link between oxidative stress and cancer, including thyroid cancer (Florian et al., 2010; Noda & Wakasugi, 2001; Wang et al., 2011). Although some information is known about the antioxidant status of patients with thyroid disorders (Erdamar et al., 2010; Senthil & Manoharan, 2004; Wang et al., 2011), comparisons between the benign and malignant groups have rarely been performed, especially on two types of benign thyroid disorders which are multinodular goitre (MNG) and follicular thyroid adenoma (FTA). Hence, this study focuses on MNG and FTA as the benign groups, while two of the most common thyroid cancers, papillary thyroid cancer (PTC) and follicular thyroid cancer (FTC) as the malignant groups. Such comparisons can provide data on the potential associations between oxidative stress and development of malignant form from the benign cases.

As the metabolic effects of thyroid hormones are linked to oxidative stress, genetic variations in antioxidant enzymes may also have an impact on the endogenous antioxidant defence system (Da Costa, Badawi & El-Sohemy, 2012; Maier et al., 2007). There is a lack of studies investigating the genetic variations of antioxidant enzymes in relation to thyroid disorder-associated oxidative stress, particularly through single nucleotide polymorphism (SNP) analyses.

Therefore, the aim of this study was to analyse the antioxidant status in patients with thyroid disorders by measuring antioxidant activities and parameters of oxidative stress including antioxidant enzyme activities, lipid peroxidation and levels of ROS. In addition, nucleotide alterations in genes encoding the antioxidant enzymes, superoxide dismutase (SOD), glutathione peroxidase (GPx), and catalase (CAT), were screened in order to assess the underlying genetic cause that may be linked to thyroid disorders. For the screening of gene alterations, whole exome sequencing (WES) analysis were performed in both MNG (n = 9) and PTC (n = 9) groups as they are the most common form of thyroid disorders. To minimise cost, gene alterations detected in the MNG and PTC patients were reanalysed in patients with FTC (n = 5); a less common form of thyroid cancer, by PCR–DNA sequencing analysis.

Materials and Methods

Subjects

A total of 41 patients with thyroid gland disorders from University of Malaya Medical Centre, Kuala Lumpur were involved in this study. Consents were obtained from all the patients before proceeding with the collection of samples. Ethical clearance for this study was obtained from the UMMC medical ethics committee (reference number: 925.8). The patients were categorised into four groups according to their diagnosis of thyroid gland disorders; MNG, FTA, PTC, and FTC. Their blood samples were taken prior to surgery, for antioxidant analyses and thyroid function tests. In addition, a total of 14 healthy subjects served as control group. Blood samples of the patients and control group were sent to Clinical Diagnostic Laboratory (CDL) of UMMC for the analyses of TFT using ADVIA Centaur CP Immunoassay System (TSH normal reference: 0.55–4.78 mU/L; fT4 normal reference: 11.5–22.7 pmol/L) (Table 1).

Table 1 Demographic and clinical characteristics of patients with their thyroid function test (TFT) results.

	Control (n = 14)	MNG (n = 18)	FTA (n = 7)	PTC (n = 10)	FTC (n = 6)	
Age (years)	32.5 (29–66)	53.5 (35–71)	34 (14–72)	49.5 (22–76)	57.5 (35–80)	
Gender	F: 11	M: 3	F: 16	M: 2	F: 6	M: 1	F: 5	M: 5	F: 6	M: 0	
TSH (mU/L)	1.24 (0.43–3.1)	1.02 (0.17–5.35)	1.98 (0.88–2.74)	1.73 (0.41–4.63)	1.13 (0.01–2.51)	
fT4 (pmol/L)	16.3 (13.6–21.5)	16.1 (12.0–20.2)	17.9 (14.2–19.4)	16.7 (13.7–19.2)	16.05 (12.4–44.5)	
Notes:

Data are presented as median. Patient groups: MNG, multinodular goitre; FTA, follicular thyroid adenoma; PTC, papillary thyroid cancer; FTC, follicular thyroid cancer. Gender: Female (F), Male (M). TSH, thyroid-stimulating hormone (normal reference: 0.55–4.78 mU/L); fT4, free thyroxine (normal reference: 11.5–22.7 pmol/L).

Meanwhile tissue samples of thyroid glands taken from patients with thyroid disorders were used for analyses of genes encoding the antioxidant enzymes. For the WES analysis, selected patients from the MNG (n = 9) and PTC (n = 9) groups were included. For comparison, genes alterations detected in the MNG and PTC patients were also analysed in the FTC patients, using PCR–DNA sequencing. Tissue samples were extracted from the nodule and their clinical diagnoses were confirmed by histopathological examination via H&E stain. The tissue samples were kept fresh-frozen prior to DNA extraction.

Blood samples

Blood (3 mL) was drawn by venepuncture from the subjects. Serum was collected into BD Vacutainer® SST tube and was centrifuged at 2,236g for 15 min at 4 °C. Whole blood was collected into BD Vacutainer® K2EDTA tube and centrifuged at 2,236g for 15 min at 4 °C to separate plasma and red blood cells (RBC). Plasma was isolated and the RBC was washed with phosphate buffered saline (PBS) three times, followed by centrifugation at 1,000g for 5 min at 4 °C. Double distilled water was added into the tube, and the RBC was allowed to lyse for 2 h at 4 °C. After the incubation, RBC was centrifuged at 2,000g for 15 min at 4 °C. The supernatant, containing lysed RBC, and the serum samples were stored at −80 °C until further analysis.

Superoxide dismutase

Superoxide dismutase enzyme activity was measured using commercially available SOD Assay Kit (Cayman, Ann Arbor, Michigan, USA) according to the manufacturer’s protocol. Serum and RBC samples were diluted to 1:5 and 1:100 with sample buffer, respectively. The measurement was based on the activity of SOD in the sample to cause dismutation of the superoxide radicals generated by xanthine oxidase and hypoxanthine. The absorbance was read at 450 nm. One unit of SOD is defined as the amount of enzyme needed to exhibit 50% dismutation of the superoxide radicals. The analyses were performed in triplicate.

Catalase

Catalase enzyme activity was measured using commercially available CAT Assay Kit (Cayman, USA) according to the manufacturer’s protocol. The assay utilised the peroxidatic function of the enzyme. Formaldehyde served as the standard, and samples were diluted with sample buffer in a ratio of 1:5 for serum, and 1:1,000 for RBC prior to the assay. The reactions were initiated with the addition of H2O2. The assay is based on the reactions of CAT with methanol in the presence of H2O2. The absorbance was read at 550 nm using a plate reader. CAT activity was expressed as unit per millilitre whereby one unit is defined as the amount of enzyme that will cause the formation of 1.0 nmol of formaldehyde per minute at 25 °C. The assays were done in triplicate.

Glutathione peroxidase

Glutathione peroxidase enzyme activity was measured using commercially available GPx Assay Kit (Cayman Chemical, Ann Arbor, MI, USA), according to the manufacturer’s protocol. Serum and RBC samples were diluted into 1:10 ratio with sample buffer. GPx activity was measured indirectly by a coupled reaction with glutathione reductase, and the reactions were initiated by the addition of cumene hydroperoxide. The absorbance was read every minute at 340 nm for 5 min using a plate reader to obtain five time points. GPx activity was expressed as unit per millilitre whereby one unit is defined as the amount of enzyme that will cause the oxidation of 1.0 nmol of NADPH to NADP+ per minute at 25 °C. All analyses were performed in triplicate.

ABTS radical scavenging activity

The 2,2′-azino-bis(3-ethylbenzothiazoline-6-sulphonic acid) (ABTS) radical scavenging activity in the blood samples was measured based on the method of Magalhães et al. (2008) with slight modification. ABTS (7 mM) and potassium persulphate (2.45 mM) were mixed together to generate ABTS+ radical cation by incubating the solution in the dark at room temperature for 16 h. The solution was then diluted with methanol to obtain an absorbance of 0.70 ± 0.02 at 415 nm. Four microliters of samples (serum or RBC) were added to 400 μL of ABTS reagent and the mixture was incubated in the dark for 10 min. After incubation, the sample was centrifuged at 1,643g for 1 min, and the absorbance of the supernatant was read at 415 nm. Trolox was used as standard and was similarly analysed as above. Trolox standard curve was plotted with a concentration range of 0, 0.5, 1, 1.5, and 2 mM. The final results were expressed as Trolox equivalent antioxidant capacity (TEAC).

ROS scavenging assay

Dichlorofluorescein diacetate (DCF-DA) was used as the fluorescence-based probe for the detection of ROS in the blood samples. In the assay, 5 μL of sample and 100 μL of DCF-DA reagent (20 μM) were added into a black 96-well plate and was incubated for 30 min at room temperature. Fluorescence reading was taken with the excitation and emission wavelengths set at 485 and 530 nm, respectively (Bioplex 200 Systems) (Bio-Rad, Hercules, CA, USA). All reactions were done in triplicate. Results were expressed as relative fluorescence unit.

Lipid peroxidation

Thiobarbituric acid reactive substances (TBARS) assay was used in estimating lipid peroxidation in the blood samples. Malondialdehyde (MDA) as the by-product of lipid peroxidation, reacted with thiobarbituric acid (TBA) to form TBARS. TBARS reagent was prepared by mixing 0.3 g of TBA, 12 g of trichloroacetic acid (TCA) and 1.04 mL of 70% perchloric acid (HClO4) in 80 mL of double-distilled water. Sample (50 μL) was mixed with 250 μL of TBARS reagent and boiled at 90 °C for 20 min. After cooling on ice, the mixture was centrifuged at 309g for 10 min at 25 °C. Absorbance of the supernatant was read at 532 nm (Bio-Rad Model 680 Microplate Reader, Hercules, CA, USA). A standard curve was constructed using 1,1,3,3-tetraethoxypropane (TEP) (0–100 ng/μL). Protein content of the sample was measured using Bradford protein assay, with bovine serum albumin (BSA) as the standard. The levels of lipid peroxidation were expressed as nanomole MDA equivalents per milligram of protein. All analyses were performed in triplicate.

Molecular analysis

DNA extraction and whole exome sequencing

DNA samples of the patients; MNG (n = 9) and PTC (n = 9), were sent to BGI Tech Solutions, Shenzhen, China for WES analysis. The protocols were based on the Agilent SureSelect XT Target Enrichment System for Illumina Paired-End Sequencing which involved the following steps; library preparation, hybridisation and capture, and sequencing. The raw image files obtained from the sequencing were processed using the Illumina Software 1.7 via base-calling, and the raw data was saved in the FASTQ format. The quality control (QC) step was performed using FastQC (www.bioinformatics.babraham.ac.uk). Data filtering was applied to the FASTQ data and the sequencing reads were aligned to the Reference Genome of human genome build37 (hg19) using BWA software (http://bio-bwa.sourceforge.net). The SOAPsnp software was used for SNP calling (http://soap.genomics.org.cn) and following identification of SNPs, an in-house AnnoDB software (AnnotationDB) was used for RefGene annotation.

SNP genotyping of FTC patients

TaqMan SNP Genotyping Assays (Applied Biosystems, Foster City, CA, USA) were used to amplify specific SNP alleles in purified DNA samples of FTC (n = 5) patients using real-time polymerase chain reaction (qRT-PCR). The list of genes with their respective SNP primers used for the qRT-PCR are listed in Table 2. qRT-PCR amplification was performed in 0.2 mL MicroAmp® Optical 8-tube strips in a final volume of 10 μL that consisted of 5 μL of 2× TaqMan® Genotyping Master Mix, 0.5 μL of TaqMan® DME Genotyping Assay Mix, and genomic DNA diluted in distilled water (10 ng). qRT-PCR was performed using the following PCR parameters; 10 min of initiation steps at 95 °C, followed by 50 cycles of denaturation at 92 °C for 15 s, and annealing steps for 90 s.

Table 2 List of genes and SNP primers of TaqMan® SNP Genotyping Assays.

Gene	dbSNP	SNP Context sequences[VIC/FAM]1	TaqMan assay ID	
SOD2	rs752779	ACTTGTTCCTATAGCATTTAAAAAG[A/G]TCTCCCTGTACCTGTGCTGCATATT	C__630044_10	
SOD3	rs2536512	CATGCAGCGGCGGGACGACGACGGC[A/G]CGCTCCACGCCGCCTGCCAGGTGCA	C__2668728_10	
GPX3	rs3763011	TTTAAGCACTTAATATTAAGTACCC[A/G]AAAAGCACTTATTAAGTGCTTTCAT	C__27513029_10	
GPX3	rs8177447	CCTCAAGCAAGGTTGACACTCCTCT[C/T]ATCCCTGCTCTAGAACTCCTGTCCT	C__30052985_10	
GPX7	rs1970951	AGCGAGACCTGGGCCCCCACCACTT[C/T]AACGTGCTCGCCTTCCCCTGCAACC	C__11730536_10	
GPX8	rs381852	GAGGGTCAAGTTGTGAAGTTCTGGA[A/G]GCCAGAGGAGCCCATTGAAGTCATC	C__9528830_10	
CAT	rs769217	CTCGAGTGGCCAACTACCAGCGTGA[C/T]GGCCCGATGTGCATGCAGGACAATC	C__3102907_10	
CAT	rs769218	ATTGCAAAGCTATGTACCCGTGACA[A/G]TGTAAATGAAAGGTTTGATTGTGCT	C__3102900_10	
Note:

1 Shows a reporter dye for the specific alleles underlined and highlighted in blue. Each assay mix concentration is 40× in 188 μL per tube. The amplicon size is NA.

Statistical analyses

All data for antioxidant analyses were expressed as mean ± SEM, unless otherwise stated. The Statistical Package for Social Sciences (SPSS) version 24.0 (IBM, New York, NY, USA) was used to perform the following statistical analyses. One-way analysis of variance (ANOVA) was used to compare the mean differences of serum and RBC antioxidant analyses between the groups of subjects for all normally distributed dataset whilst the non-parametric Kruskal–Wallis test was used, if otherwise. The Tukey post-hoc test was used for multiple comparisons of specific sample pairs when equal variances were assumed whilst Dunnett’s T3 post-hoc test was used when unequal variances were assumed. Levene’s test of homogeneity of variances was used to verify the assumption. Spearman’s correlation coefficient test was used to compare the association between SNPs and antioxidant enzyme activities among MNG and PTC groups of subjects.

Hardy–Weinberg (HW) exact test was used to test the genetic equilibrium of the SNPs using Genepop (version 4.2) population genetics software package (Raymond & Rousset, 1995; Rousset, 2008). The distribution of alleles and genotypes of SNPs between MNG, PTC, and FTC were analysed using Fisher’s exact test. p-values less than 0.05 were considered significant.

Results

Table 1 shows the demographic and clinical characteristics of patients, together with their thyroid function tests. The patients’ age ranged from 14 to 80 years old while the controls ranged from 29 to 66 years old. Only two patients were diagnosed with thyroiditis, implying that thyroiditis is not common among the recruited patients. There was no significant difference between the TSH concentration and fT4 values between the control group and the thyroid disorder groups (p > 0.05).

Antioxidant response and oxidative stress in serum

Figures 1A–1C shows the antioxidant enzyme activities in serum of control and patients. Patients from the MNG and PTC groups had significantly higher levels of SOD activities (more than fourfold) compared to control (Fig. 1A). Out of the four thyroid disorder groups, only MNG showed higher CAT activities than control (Fig. 1B). Meanwhile, there were no significant differences in GPx activities between control subjects and the thyroid disorder groups (Fig. 1C).

Figure 1 Antioxidant analysis in serum samples of thyroid patients.

1a: SOD activity; 1b: CAT activity; 1c: GPx activity; 1d: ABTS radical scavenging activity, expressed as Trolox Equivalent Antioxidant Activity (TEAC); 1e: lipid peroxidation assay measured as MDA concentration; 1f: ROS analysis. # Indicates significant difference (p < 0.001) between the diseased groups of benign and malignant thyroid disorders (MNG vs PTC, FTA vs FTC). * indicates significant difference (p < 0.05 and > 0.01) between control and the diseased groups. ** indicates significant difference (p < 0.01 and > 0.001) between control and the diseased groups. *** indicates significant difference (p < 0.001) between control and the diseased groups.

Analysis of ABTS radical scavenging activity shows the thyroid disorder groups to have significantly lower antioxidant activities than the control group (p < 0.05) (Fig. 1D). Amongst the four diseased groups, PTC showed the lowest ABTS radical scavenging activity which was significantly lower than its benign form, MNG. Lipid peroxidation was estimated by measuring levels of MDA, a by-product of lipid peroxidation. Levels of MDA equivalents were found to be significantly increased in the FTA group, compared to control while there was not significant difference in the remaining groups (Fig. 1E). Another parameter for oxidative stress that was measured was levels of ROS (Fig. 1F). ROS level was significantly higher in the PTC group, compared to control while there were no changes in the remaining groups.

Antioxidant response and oxidative stress in RBC

Figures 2A–2C shows the antioxidant enzyme activities in RBC of control and patients. Patients with MNG and FTC had significantly lower SOD activities than control (Fig. 2A). Amongst the four thyroid disorder groups, the FTA group showed lower CAT activities than control (Fig. 2B). GPx activities of the malignant groups (PTC and FTC) were significantly lower than the benign (MNG and FTA) and control groups (p < 0.05). On the contrary, there were no significant changes in the MNG and FTA groups when compared to control (p > 0.05) (Fig. 2C).

Figure 2 Antioxidant analysis in red blood cells samples of thyroid patients.

(A) SOD activity; (B) CAT activity; (C) GPx activity; (D) ABTS radical scavenging activity, expressed as Trolox Equivalent Antioxidant Activity (TEAC); (E) lipid peroxidation assay measured as MDA concentration; (F) ROS analysis. # Indicates significant difference (p < 0.001) between the diseased groups of benign and malignant thyroid disorders (MNG vs PTC, FTA vs FTC). * indicates significant difference (p < 0.05 and > 0.01) between control and the diseased groups. ** indicates significant difference (p < 0.01 and > 0.001) between control and the diseased groups. *** indicates significant difference (p < 0.001) between control and the diseased groups.

Analysis of ABTS radical scavenging activities indicated no significant changes among the thyroid disorder groups with that of control (Fig. 2D). However, levels of MDA equivalents in MNG and PTC groups were significantly higher than the control group (Fig. 2E) while only the PTC group showed significantly higher level of ROS compared to control (Fig. 2F).

Single nucleotide polymorphisms in patients

Whole exome sequencing analysis of MNG (n = 9) and PTC (n = 9) revealed 11 SNPs in SOD gene, 28 SNPs in GPX gene, and 10 SNPs in CAT gene (Table 3). Out of the total 49 SNPs identified from the sequencing, eight SNPs were observed to be commonly present in both groups of thyroid disorders and in five of the FTC patients (Table 4). Thus, Venn diagrams (Fig. 3) were constructed to show the distribution of these SNPs in all the three groups; MNG, PTC, and FTC. These diagrams provide information on patients sharing similar polymorphisms among the three types of thyroid disorders groups. The most common SNP for SOD gene is rs752779, detected in all MNG patients (n = 9), eight PTC patients and in all FTC patients (n = 5). Meanwhile rs2536512 was detected in six PTC patients and only in three of MNG and FTC patients. For the GPX gene, four SNPs are found to be common; rs1970951, rs381852, rs3763011, and rs8177447. Eight MNG and all PTC patients appeared to have rs1970951 and rs381852, whereas the former and the latter SNPs was detected in four and three of the FTC patients, respectively. Four FTC and seven patients from each of the MNG and PTC groups have rs3763011. The same FTC and MNG patients as well as all PTC patients have rs8177447. The CAT SNPs; rs769217 and rs769218 were shared by the same patients of MNG (n = 4) and PTC (n = 7). Meanwhile all FTC patients (n = 5) have rs769217, and only four FTC patients have rs769218 (Fig. 3).

Table 3 Characteristics and localisations of SNPs in SOD2, SOD3, GPX1, GPX2, GPX3, GPX4, GPX6, GPX7, GPX8, and CAT genes in patients with MNG and PTC.

Gene	dbSNP	Location	Types of variants	Base; amino acid change	
SOD2	rs4880	Exon 2	Missense	A/G; V16A	
Ch6	rs752779	Intergenic	–	G/A;–	
q25.3	rs2758332	Intron 3	–	C/A;–	
rs2855116	Intron 3	–	A/C;–	
rs2070994	Intron 3	–	A/T;–	
rs2842960	Intron 2	–	C/T;–	
SOD3	rs8192291	Exon 2	Synonymous	C/T; L71L	
Ch4	rs1799895	Exon 2	Missense	C/G; R231G	
p15.2	rs2695232	UTR3′	–	C/T;–	
rs2855262	UTR3′	–	T/C;–	
rs2536512	Exon 2	Missense	G/A; A58T	
GPX1	rs368623389	Exon 1	Missense	T/C; E89G	
Ch3	
p21.3	
GPX2	rs17881414	Intron 1	–	T/C;–	
Ch14	
q23.3	
GPX3	rs11548	UTR3′	–	C/T;–	
Ch5	rs2070593	UTR3′	–	G/A;–	
q33.1	rs2277940	UTR3′	–	T/C;–	
rs8177413	Exon 1	Synonymous	G/C; L13L	
rs2230303	UTR3′	–	T/G;–	
rs8177448	UTR3′	–	G/A;–	
rs8177412	UTR5′	–	T/C;–	
rs3763012	Intron 1	–	G/A;–	
rs3763011	Intron 1	–	G/A;–	
rs869975	Intron 2	–	G/A;–	
rs8177447	Intron 4	–	T/A;–	
rs870407	Intron 1	–	A/G;–	
rs870406	Intron 1	–	G/A;–	
rs869976	Intron 2	–	A/G;–	
GPX4	rs713041	UTR3′	–	T/C;–	
Ch19	rs4807542	Exon 1	Synonymous	G/A; P12P	
p13.3	rs569667691	Exon 7	Synonymous	G/A; L193L	
rs8178977	Intron 6	–	G/C;–	
GPX6	rs372898956	Exon 4	Synonymous	C/T; V122V	
Ch6	rs974334	Intron 2	–	C/G;–	
p22.1					
GPX7	rs1970951	Exon 2	Synonymous	T/C; F79F	
Ch1	rs11810754	UTR5′	–	T/G;–	
p32	rs1970950	Intron 2	–	A/G;–	
GPX8	rs381852	Exon 3	Missense	A/G; K182R	
Ch5	rs10059517	UTR3′	–	T/C;–	
q11.2	rs2270910	Intron 1	–	C/G;–	
CAT	rs769217	Exon 9	Synonymous	C/T; D389D	
Ch11	rs704724	Exon 10	Synonymous	C/T; L419L	
p13	rs1049982	UTR5′	–	T/C;–	
rs7943316	UTR5′	–	A/T;–	
rs10836235	Intron 1	–	C/T;–	
rs769218	Intron 1	–	G/A;–	
rs17881488	Intron 11	–	C/A,–	
rs12270780	Intron 1	–	G/A;–	
rs2073058	Intron 7	–	A/G;–	
rs16925614	Intron 11	–	C/T;–	
Note:

V16A, Valine16Alanine; L71L, Leucine71Leucine; R231G, Arginine231Glycine; A58T, Alanine58Threonine; E89G, Glutamic acid89Glycine; L13L, Leucine13Leucine; P12P, Proline12Proline; L193L, Leucine193Leucine; V122V, Valine122Valine; F79F, Phenylalanine79Phenylalanine; K182R, Lysine182Arginine; D389D, Aspartic acid389Aspartic acid; L419L, Leucine419Leucine.

Table 4 Genotype and allele frequencies of the eight SNPs in MNG, PTC, and FTC patients.

SNPs	Genotype	Number of patients (%)1	p-value	Allele	Total haplotypes (%)2	p-value	
MNG (n = 9)	PTC (n = 9)	FTC (n = 5)	MNG	PTC	FTC	
rs752779	GG	4	(44)	1	(11)	4	(80)	*0.03	G	13	(72)	9	(50)	9	(90)	0.08	
GA	5	(56)	7	(78)	1	(20)	A	5	(28)	9	(50)	1	(10)	
AA	0	0	1	(11)	0	0								
rs2536512	GG	0	0	1	(11)	2	(40)	0.22	G	3	(17)	7	(39)	5	(50)	0.14	
GA	3	(33)	5	(56)	1	(20)	A	15	(83)	11	(61)	5	(50)	
AA	6	(67)	3	(33)	2	(40)								
rs1970951	TT	6	(67)	7	(78)	0	(0)	*0.05	T	14	(78)	16	(89)	4	(40)	*0.03	
TC	2	(22)	2	(22)	4	(80)	C	4	(22)	2	(11)	6	(60)	
CC	1	(11)	0	(0)	1	(20)								
rs381852	AA	4	(45)	5	(56)	1	(20)	0.32	A	12	(67)	13	(72)	4	(40)	0.25	
AG	4	(45)	3	(33)	2	(40)	G	6	(33)	5	(28)	6	(60)	
GG	1	(10)	1	(11)	2	(40)								
rs3763011	GG	1	(11)	2	(22)	0	(0)	0.91	G	8	(44)	9	(50)	4	(40)	0.93	
GA	6	(67)	5	(56)	4	(80)	A	10	(56)	9	(50)	6	(60)	
AA	2	(22)	2	(22)	1	(20)								
rs8177447	TT	7	(78)	8	(89)	3	(60)	0.55	T	15	(83)	17	(94)	7	(70)	0.25	
TC	1	(11)	1	(11)	1	(20)	C	3	(17)	1	(6)	3	(30)	
CC	1	(11)	0	(0)	1	(20)									
rs769217	CC	2	(22)	3	(34)	1	(20)	0.42	C	6	(33)	10	(56)	6	(60)	0.36	
CT	2	(22)	4	(44)	4	(80)	T	12	(67)	8	(44)	4	(40)	
TT	5	(56)	2	(22)	0	(0)					
rs769218	GG	2	(22)	3	(34)	0	(0)	0.54	G	6	(33)	10	(56)	4	(40)	0.48	
GA	2	(22)	4	(44)	4	(80)	A	12	(67)	8	(44)	6	(60)	
AA	5	(56)	2	(22)	1	(20)								
Note:

1Genotype frequencies and 2allele frequencies were determined by Fisher’s exact test analysis and *p < 0.05 was considered statistically significant between the three groups. All genotypes in MNG, PTC, and FTC populations were in associations with HW (p > 0.05).

Figure 3 Venn diagrams of SNPs distribution among patients of MNG (M1–M9), PTC (P1–P9), and FTC (F1–F5).

Individual patients in each group; MNG (M1–M9), PTC (P1–P9), and FTC (F1–F5).

Analysis on genotypic and allelic frequencies

Genotypic and allelic frequencies in MNG, PTC, and FTC patients of the eight SNPs are summarised in Table 4. The HW Exact test showed that the genotype and allele frequencies in all SNPs conformed to HW equilibrium (p > 0.05). The allelic frequency distribution of the eight SNPs (HW—p > 0.05) in our studied populations was then compared with the healthy subjects of South Asian (SAS) and East Asian (EAS) countries, taken from the database of 1000 Genomes Project Phase 3 allele frequencies (Yates et al., 2016). The populations were categorised as SAS, EAS, MNG, PTC, and FTC, and the data are presented in Fig. 4 according to their allele frequencies.

Figure 4 Allelic frequencies distribution of the highlighted eight SNPs in SOD, GPX, and CAT genes in our studied populations with comparison to the healthy individuals of South Asian (SAS) and East Asian (EAS).

The data of SAS and EAS populations were obtained from 1000 Genomes Project Phase 3 allele frequencies (http://Ensembl.org).

Significant differences were detected in the genotype frequencies for two SNPs; rs752779 (p = 0.03) and rs1970951 (p = 0.05) among the MNG, PTC, and FTC groups (Table 4). For the rs752779, the majority of patients with PTC have GA genotype (78%) whilst patients with FTC have GG genotype (80%). Meanwhile, for the rs1970951, the majority of patients with MNG and PTC have TT genotype (67% and 78%, respectively) whilst patients with FTC have TC genotype (80%). A significant difference in frequency was also observed for the T/C allele for the rs1970951 SNP but not for the G/A allele in the rs752779 SNP. All of the other SNPs did not show significant differences in both genotypic and allelic frequencies.

Relationship between serum enzyme activities and total SNPs identified

The overall results of the total 49 SNPs for the antioxidant enzymes were compared with the respective antioxidant enzyme activities to ascertain if any correlation exists between these two factors, in MNG and PTC patients. The analysis did not demonstrate any correlation between total SNPs and the corresponding antioxidant enzyme activities in the two groups (Fig. 5).

Figure 5 Associations between antioxidant enzyme activities and the total number of SNPs present in MNG and PTC groups.

(A) SNPs of SOD and SOD activities in MNG and PTC patients; (B) SNPs of GPx and GPx activities in MNG and PTC patients: (C) SNPs of CAT and CAT activities in MNG and PTC patients. Data are presented as mean ± SEM. Spearman’s correlation coefficient test were expressed as r for the total number of SNPs in each patients’ group, where applies.

Discussion

Thyroid disorders are more common among females with an estimated incidence of 2%, whereas it is only 0.2% in males. In fact, being female constitute one of the risk factors for thyroid disorders (Eugène, Djemli & Van Vliet, 2005; James & Kumar, 2012). The incidence of thyroid disorders was also reported to increase with age (Sapini, Rokiah & Nor Zuraida, 2009). Our study also showed a higher proportion of female patients compared to males.

Several studies have been conducted on the effects of thyroid hormones on antioxidant status and oxidative stress (Babu, Jayaraaj & Prabhakar, 2011; Cano Europa, Margarita & Rocio, 2012; Mancini et al., 2013). Although a majority of studies have reported increased oxidative stress being positively correlated with thyroid disorders, some have also reported the opposite or no changes (Venditti et al., 1997; Villanueva, Alva-Sanchez & Pacheco-Rosado, 2013). Studies investigating antioxidant status in thyroid diseases may not include all the crucial parameters to ascertain the relationship between thyroid diseases and antioxidants. Most of the previous studies regarding antioxidant status in thyroid diseases were performed either by evaluating antioxidants as a whole (Sivakanesan, Wickramarathne & Nanayakkara, 2014; Wang et al., 2011), or by measuring each antioxidants separately (Cetin et al., 2015; Erdamar et al., 2010; Senthil & Manoharan, 2004). In order to overcome this limitation, in this study, we performed analyses to determine both the antioxidant levels as well as parameters of oxidative stress in patients with thyroid disorders to provide an overall picture with regards to the status of antioxidants and oxidative damage.

We also compared these indicators between the benign and malignant forms of thyroid disorders to ascertain their potential influence in the development of cancers from the benign forms. Furthermore, we have also analysed the RBC to ascertain the effects of the thyroid disorders on the antioxidant status of RBC. Previous reports have also used thyroid tissues, however, due to difficulty in obtaining thyroid tissue samples, comparisons were made between our results and other published results of thyroid tissues. This will, at the same time, provide data on potential clinical application of serum in thyroid disorders.

The 2,2′-Azino-bis(3-ethylbenzothiazoline-6-sulphonic acid) assay measures the overall antioxidant capacity in the samples. The assay relies on the ability of antioxidants in the serum sample to scavenge ABTS radicals and inhibit oxidation in comparison to Trolox as the standard (Magalhães et al., 2008). Serum may contain some low molecular weight compounds with antioxidant activities such as vitamin C, vitamin E, and uric acid. Results from both serum and RBC samples in this study generally indicated patients with thyroid diseases tend to have lower antioxidant activities in the serum but no changes in RBC, compared to healthy subjects. A study by Wang et al. (2011) that measured antioxidant status of thyroid cancer patients demonstrated similar results as ours. Majority of studies have reported on the lowering of antioxidant activities with hyperthyroidism (Andryskowski & Owczarek, 2007; Asayama et al., 1989; Venditti, De Rosa & Di Meo, 2003) although one study has reported no significant changes (Sivakanesan, Wickramarathne & Nanayakkara, 2014). The lower antioxidant activities were accompanied with increased lipid peroxidation in serum of FTA group and increased ROS in serum of PTC group, implying reduced ability to protect against oxidative stress in these groups. Similar observation was seen for RBCs. PTC has the highest serum ROS concentration among the studied group, possibly due to the substantial reduction in serum antioxidant activities. When comparing the benign and malignant forms, there was significant reduction in serum antioxidant activities which was accompanied with increased ROS, in PTC group compared to MNG. The same could not be seen between FTA and the malignant form, FTC. The malignancy of the thyroid gland could have led to generation of high amounts of ROS, triggering oxidative stress which in turns lowers the amount of antioxidants.

A study has reported increased MDA levels in thyroid tissues of PTC patients compared to non-cancerous tissues (Erdamar et al., 2010). ROS consist of superoxide anion (O2−), hydroxyl radicals (OH−), nitric oxide (NO), and peroxyl radical (ROO−). Their production may be influenced by antioxidant enzymes including SOD, CAT, and GPx. SOD enzyme causes dismutation of superoxide anion radical, leading to production of another radical, hydrogen peroxide (H2O2). CAT and GPx convert H2O2 into oxygen and H2O (MatÉs, Pérez-Gómez & De Castro, 1999).

In addition to endogenous antioxidants, the body protects against oxidative damage through antioxidant enzymes. Studies have shown that oxidative stress can affect levels of antioxidant enzymes (Gerenova & Gadjeva, 2005; Sivakanesan, Wickramarathne & Nanayakkara, 2014; Villanueva, Alva-Sanchez & Pacheco-Rosado, 2013), hence measuring their levels in this study can provide an indication of oxidative stress in the patients. The increase in serum SOD and CAT activities in some of the thyroid disorder groups indicated increased production of superoxide anion and hydrogen peroxide, respectively. Superoxide anions are produced as by-products of the mitochondrial oxidation system. As thyroid hormones also function to regulate the mitochondrial oxidation system, disorders of the thyroid gland may have altered oxidation, leading to increased production of superoxide anion. Similarly, reaction of SOD in removing superoxide anions leads to the production of H2O2 and this explains the increased CAT activities. The increased SOD and CAT activities could also be due to up-regulation in the synthesis of these enzymes as a result of the thyroid disorders, as protective response against oxidative stress. However, despite the increased SOD and CAT activities, MDA and ROS levels were still elevated in some groups. This indicates the inability of the cells to completely scavenge the radicals and implies that the rate of clearance of the ROS by these two enzymes is slower than their production. Hyperthyroid rats were reported to have significantly higher lipid peroxidation, SOD, CAT, and GPx in liver, together with lower serum antioxidant status (Messarah et al., 2010). It was recently reported that changes in lipid peroxidation and SOD between normal and PTC thyroid tissues can be used as possible markers to differentiate between malignant and non-malignant thyroid tumours (Stanley et al., 2016).

Prolonged excessive production and inadequate removal of ROS can create an oxidative stress environment, leading to DNA damage which can contribute to the pathogenesis of the malignant state. It was recently reported that PRDX1 and PRDX6 expressions were significantly reduced in all PTCs compared to normal tissues, via V600E mutation in the BRAF gene (Nicolussi et al., 2014). PRDXs play several roles in the body including catalysing the reduction of H2O2. In a recent case report of a patient with concurrent benign thyroid cyst and PTC, based on WES and mutation analyses, it was hypothesised that prolonged H2O2 insult could interfere with the MAPK and STAT3 pathways, causing malignant transformation of the benign thyroid nodule (Lee et al., 2016).

Sadani & Nadkarni (1996) reported increased MDA levels and SOD activities in thyroid tissues of FTA, FTC, and PTC patients but no changes in the MNG group. Some of these observations are similar to what we observed in serum of the patients, implying that the changes seen in the serum could be the result of damage to the thyroid tissues. Also, higher oxidative stress in the serum is indicative of tissue damage due to oxidation, potentially damage from the thyroid tissues themselves. Indeed, studies have reported reduced antioxidant activities and increased ROS production in thyroid glands of patients with thyroid disorders (Sarkar et al., 2006; Venditti, De Rosa & Di Meo, 2003; Villanueva, Alva-Sanchez & Pacheco-Rosado, 2013).

Results for antioxidant activities, lipid peroxidation and ROS levels obtained in the RBC were similar to serum, in some instances, while for others, were different. SOD activity of RBC was lower than control. One interesting observation from the results of the RBC is the substantial reduction of GPx activities in both malignant forms of the thyroid disorder, PTC and FTC. Asayama et al. (1989) and Fernández & Videla (1989) have reported significant reduction in GPx activity of rats suffering from hyperthyroidism.

A study reported increased antioxidant enzyme activities in erythrocytes of toxic MNG patients, which is the opposite of what we observed in RBC (Alicigüzel et al., 2001). Another study also reported increased GPx in both hyper- and hypo-thyroid rats but no changes in SOD and CAT (Araujo et al., 2011). It has been reported that erythrocytes of patients/animals with hyperthyroidism showed increased oxygen consumption (Alicigüzel et al., 2001), while Sarkar et al. (2006) also reported the same observation in peripheral blood mononuclear cells (PMC). This may potentially lead to increased production of ROS, hence altering the redox balance of erythrocytes, in favour of oxidative stress.

Results from our study seem to indicate higher oxidative stress in the serum as opposed to RBC. We speculated that RBC may contain antioxidant response system that is better able to counter the effects of increased ROS production as a result of thyroid disorders. Erythrocytes are exposed to endogenous and exogenous sources of ROS including superoxide anion and H2O2. However, they are equipped with antioxidant system comprising of non-enzymatic and enzymatic antioxidants, the latter includes CAT, SOD, and GPx. The especially low levels of GPx in PTC and FTC groups indicated the inability of erythrocytes to scavenge H2O2, which is depicted by the increase in ROS and MDA levels. The reduction seen in both the malignant forms implies the possibility of the tumour affecting GPx activities or vice versa. Deficiency of GPx has been associated with several conditions. Cells/tissues deficient in GPx are more susceptible to oxidative damage, apoptosis, cell injury, and cell death (Flentjar et al., 2002; Van Remmen et al., 2004). The substantial reduction in GPx activities in the malignant form of thyroid disorders can be further explored with the aim of potentially using this enzyme as a biomarker.

As antioxidant enzymes evolve to maintain cellular redox homeostasis, their enzymatic activities may be influenced either directly or indirectly by the respective genes that encode them (Gelain et al., 2009). SOD enzyme is encoded by three different isoforms; SOD1, SOD2, and SOD3. Both SOD1 and SOD3 contain copper and zinc in their catalytic centre but are localised differently; SOD1 is present in the cytosol (CuZn-SOD), while SOD3 is found in the extracellular elements (EC-SOD). SOD2 has manganese as a cofactor and is localised in the mitochondria (Mn-SOD) (Zelko, Mariani & Folz, 2002). GPx enzyme is encoded by eight different isoforms (GPX1–GPX8) which are localised mostly in the cytosol, nucleus, and extracellular space. CAT enzyme, encoded by CAT gene, is localised in peroxisomes (Margis et al., 2008). Whole exome sequencing analysis resulted in identification of SNPs present in the antioxidant enzyme genes of all 18 patients (MNG and PTC).

Whole exome sequencing analysis resulted in identification of SNPs present in the antioxidant enzyme genes of all 18 patients (MNG and PTC). rs752779 was the most common SNP detected in the patients in this study. This SNP is located at the intergenic region of SOD2 and WTAP genes of chromosome 6q25.3. To date, there is no reported study on the association of this SNP with disease development. rs2536512 is the SNP of SOD3 gene that is located on chromosome 4p15.2. It is a missense variant identified in exon 2 of SOD3 with the base change of G to A, and amino acid substitution of Alanine to Threonine (A58T). This SNP was reported to be associated with type 2 diabetes and hypertension (Naganuma et al., 2008). A previous study has also reported the association of rs2536512 SNP with significantly greater incidence of breast carcinoma (Hubackova et al., 2012). Although the rs2536512 SNP is non-functional, it was hypothesised that this SNP may act as a genetic marker for susceptibility to type 2 diabetes and hypertension (Naganuma et al., 2008; Tamai et al., 2006).

The rs1970951 SNP is a synonymous variant identified in exon 2 of GPX7 gene with a base change of T to C, maintaining the amino acid phenylalanine (F79F). There is no known link between this SNP with any diseases. rs381852 is a missense variant identified in exon 3 of GPX8 gene with the base change of A to G, and amino acid substitution of Lysine to Arginine (K182R). Meanwhile, both rs3763011 and rs8177447 are from GPX3 gene located at intron 1 and intron 4, respectively. CAT is the only enzyme without isoforms, and is encoded by the CAT gene. rs769217 is a synonymous variant in exon 9 of CAT gene with base change of C to T without the substitution of Aspartic acid, while C6 rs769218 is located at intron 1. There is also no known link between this SNP with any diseases.

A Spearman’s correlation coefficient was performed to investigate if an association exists between the antioxidant enzyme activities and SNPs in the genes encoding the antioxidant enzymes, in the thyroid disorder patients. Based on results of the SOD enzyme, the malignant group has a lower number of SNPs (n = 49) compared to the benign group (n = 55) (Fig. 5A). However, for GPX and CAT (Figs. 5B and 5C), both showed a higher number of SNPs for the malignant groups compared to the benign groups. Nevertheless, there was no significant correlation observed. This indicates that the SNPs for SOD, GPx, and CAT do not appear to influence the corresponding enzyme activities. Interestingly, although not significant, there appeared to be a trend of negative association between SNPs of CAT and GPX with the respective antioxidant enzyme activities, for the benign group whilst a positive association was observed for the malignant group. Further studies using larger sample size could validate this observation. A particular SNP may indicate that one form of the gene is common in some people with that particular trait, which means that the incidence of having certain allele frequency may be linked with the history of the trait rather than affecting the body system mechanisms.

Conclusion

This study demonstrates that patients with thyroid disorders have lower antioxidant defence system, potentially predisposing them to oxidative stress. Additionally, different forms of thyroid disorders could influence the levels of antioxidant status in the serum and RBC of these patients, implying varying capability of preventing oxidative stress. No significant association was observed between serum enzyme activities and the total number of SNPs identified in this study. SNP-disease association study however, demonstrated that the genotypic (and allelic) distributions of rs752779 and rs1970951 are markedly different among subjects with malignant thyroid disorders. The lack of association between other SNPs and thyroid disorders need to be validated in a larger sample size using appropriate controls subjects.

Supplemental Information

Supplemental Information 1 Analyses of antioxidant activities and parameters of oxidative stress in serum.

Click here for additional data file.

Supplemental Information 2 Analyses of antioxidant activities and parameters of oxidative stress in RBC.

Click here for additional data file.

Supplemental Information 3 Whole Exome Sequencing datasets for MNG and PTC patients.

Click here for additional data file.

Supplemental Information 4 Clinical data of thyroid disorder patients (MNG, PTC, FTC).

Click here for additional data file.

We would like to thank all the subjects for their participation.

Additional Information and Declarations

Competing Interests

Author Contributions

Human Ethics

Data Availability

The authors declare that they have no competing interests.

Nur Siti Fatimah Ramli conceived and designed the experiments, performed the experiments, analysed the data, wrote the paper, prepared figures and/or tables, reviewed drafts of the paper.

Sarni Mat Junit conceived and designed the experiments, contributed reagents/materials/analysis tools, wrote the paper, reviewed drafts of the paper.

Ng Khoon Leong contributed reagents/materials/analysis tools, reviewed drafts of the paper.

Nurhanani Razali analysed the data, reviewed drafts of the paper.

Jaime Jacqueline Jayapalan analysed the data, reviewed drafts of the paper.

Azlina Abdul Aziz conceived and designed the experiments, contributed reagents/materials/analysis tools, wrote the paper, reviewed drafts of the paper.

The following information was supplied relating to ethical approvals (i.e. approving body and any reference numbers):

Ethical clearance for this study was obtained from the Medical Ethics Committee of University of Malaya Medical Centre (reference number, 925.8).

The following information was supplied regarding data availability:

The raw data has been supplied as Supplemental Dataset Files.

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
