# Peer review of "Analyses of antioxidant status and nucleotide alterations in genes encoding antioxidant enzymes in patients with benign and malignant thyroid disorders"

_PeerJ, doi:10.7717/peerj.3365_

## Round 0.1 · original submission · Major Revisions

· Academic Editor

Major Revisions

Please make a revision to your manuscript strictly based upon the review comments returned from the reviewers.

Reviewer 1 ·

Basic reporting

The aim of this study was to evaluate antioxidant activities and oxidative stress markers in serum and red blood cells of patients with malignant (PTC and FTC) or benign (MNG and FTA) thyroid disease and to evaluate if any alteration was associated with specific SNPs in SODs, GPXs, and CAT. The authors found alterations of antioxidant enzymes and oxidative stress markers in serum samples and red blood cells of patients with thyroid diseases as compared to those of healthy controls. Whole exome sequencing and genotyping analysis were performed in thyroid tissues of MNG, PTC, and FTC and led to the identification of 8 SNPs commonly present in all thyroid diseases. However, no correlation was found between dysregulated activities of antioxidant enzymes and the total number of SNPs identified either in patients with benign thyroid disorders or in patients with malignant thyroid disorders.
It is a interesting study, although several concerns need to be addressed as described below.
1. I suggest a professional English editing service, because in some sentences the language could be more clear and precise. For instance, please correct the improper use of the word “mutation”. Single Nucleotide Polymorphisms (SNPs) and DNA mutations are defined as DNA variants detectable in >1 % or <1 % of the population, respectively. Since all the genetic variants identified in this study are common in both South Asian and East Asian populations, the authors should define them as “SNPs” or simply “variants”. Finally, please correct “mutation in BRAF V600E gene” (Page 13, Line 431) with “V600E mutation in the BRAF gene”.
2. Discussion section should be more concise and report exclusively comments regarding the findings of the present study or related literature.

Experimental design

3. Table 1. Add more clinical data of the study population, such as the presence or absence of thyroiditis, levels of anti-thyroid antibodies (e.g. anti-TPO and anti-TG), tumor size, tumor stage, TNM, extrathyroid extension, presence or absence of metastases. Moreover, clinical data reported in the manuscript (Page 8, Lines 243-244) are different from those reported in Table 1 and confounding. Finally, I suggest to report all scalar variables as median (minimum-maximum), because it would be more informative.
4. Sample collection: Have blood samples been collected prior or after surgery?
5. Sample collection: Add more information about tissue samples analyzed (e.g. fresh-frozen or FFPE, tumor or normal, if normal adjacent or contralateral).
6. Genetic analyses on tissue samples: Define the rationale of performing these analyses in tissues rather than other samples. Did the authors aim to identify somatic variations?
7. Genetic analyses on tissue samples: Why did the authors analyze only MNG, PTC, FTC and no FTA? Which inclusion criteria have been used?
8. Whole exome sequencing: the authors should provide detailed information of protocol used.

Validity of the findings

9. Raw data of MDA levels in serum samples: Please, check SEM values equal to zero.
10. Statistical Analyses: Student’s t test can be applied between groups that have normal distribution of data. If not, non-parametric methods should be used. Furthermore, when more than two sample groups are compared authors should use kruskal wallis test (non parametric) or ANOVA test (parametric) followed by post-hoc tests for multiple comparisons of specific sample pairs.
11. Statistical Analyses: I suggest to report p-values in figures and in the manuscript specifying range values (e.g. *, < 0.05 and > 0.01; **, <0.01 and > 0.001; ***, < 0.001) and single values, respectively.

·

Basic reporting

The work is clear and professional presented. The article is well structured.

Experimental design

The design of this work is good. The research question is well defined. References may need to be corrected.

Validity of the findings

The data is not robust, the conclusion is not clearly made.

Additional comments

In the manuscript “Analyses of antioxidant status and nucleotide alterations in genes encoding antioxidant enzymes in patients with benign and malignant thyroid disorders”, the authors detected the antioxidant activities, antioxidant enzymes and biomarkers of oxidative stress in patients with thyroid disorders. Alterations in genes encoding the antioxidant enzymes were analyzed using whole exome sequencing and PCR-DNA sequencing. This work is well done and the story is interesting. It can accepted for publication directly. The subtitles of the Results should directly present the result, but not the experiments performed. This will make it easy to read for readers. This can be made after acceptance.

Reviewer 3 ·

Basic reporting

The results section ( lines 48-54) is confusing and need to be reworded

Experimental design

Yes

Validity of the findings

The statistics does not seem to match the findings in several places

Additional comments

The article in which the authors examine the antioxidant status and nucleotide alterations in genes encoding the antioxidant enzymes in thyroid cancer patients is an interesting finding. But, addressing the following questions will improve the manuscript:
1. The results part of the abstract and the initial results section is very confusing and rewording will make it more clear.
2. Both in figure 1 and 2 the data in which the authors have seen a huge difference in values, especially for CAT activity for FTC with very low standard error is not statistically significant whereas very little differences in values for MDA is significant is intriguing.
3. Under genomic analysis Lines 293 to 296 the authors discuss about the validation experiment, but the results obtained was not discussed.

---

## Round 0.2 · accepted · Accept

· Academic Editor

Accept

Your submission is now acceptable for publication in PeerJ

Reviewer 1 ·

Basic reporting

The Authors addressed all comments. No further comments.

Experimental design

The Authors addressed all comments. No further comments.

Validity of the findings

The Authors addressed all comments. No further comments.

·

Basic reporting

The written through the manuscript is clear and the literature references are sufficient.

Experimental design

The design of the experiments are good

Validity of the findings

The findings is interesting and novel. The data is robust.

Additional comments

The manuscript is further improved after revision. It can be accepted for publication

Reviewer 3 ·

Basic reporting

No Comment

Experimental design

No Comment

Validity of the findings

No Comment

Additional comments

The authors have adequately addressed the concerns raised by the reviewers.